# Neutrophil Activation by Mineral Microparticles Coated with Methylglyoxal-Glycated Albumin

**DOI:** 10.3390/ijms23147840

**Published:** 2022-07-16

**Authors:** Elena V. Mikhalchik, Victor A. Ivanov, Irina V. Borodina, Olga V. Pobeguts, Igor P. Smirnov, Irina V. Gorudko, Daria V. Grigorieva, Olga P. Boychenko, Alexander P. Moskalets, Dmitry V. Klinov, Oleg M. Panasenko, Luboff Y. Filatova, Ekaterina A. Kirzhanova, Nadezhda G. Balabushevich

**Affiliations:** 1Federal Research and Clinical Center of Physical-Chemical Medicine, Federal Medical Biological Agency, 119435 Moscow, Russia; vanov.va@inbox.ru (V.A.I.); irabor00@gmail.com (I.V.B.); nikitishena@mail.ru (O.V.P.); smirnov_i@hotmail.com (I.P.S.); olya.lyubovskaya@gmail.com (O.P.B.); a.p.moscalets@gmail.com (A.P.M.); klinov.dmitry@mail.ru (D.V.K.); o-panas@mail.ru (O.M.P.); 2Department of Biophysics, Belarusian State University, 220030 Minsk, Belarus; irinagorudko@gmail.com (I.V.G.); dargr@tut.by (D.V.G.); 3Faculty of Chemistry, Lomonosov Moscow State University, 119991 Moscow, Russia; luboff.filatova@gmail.com (L.Y.F.); kharenko.kate@gmail.com (E.A.K.); nbalab2008@gmail.com (N.G.B.); 4Laboratory of Biomaterials, Sirius University of Science and Technology, 354340 Sochi, Russia; 5Research and Educational Resource Center for Immunophenotyping, Digital Spatial Profiling and Ultrastructural Analysis Innovative Technologies, Peoples’ Friendship University of Russia, 117198 Moscow, Russia

**Keywords:** vaterite, neutrophils, chemiluminescence, advanced glycation end-products, albumin, diabetes, methylglyoxal, hyperglycemia

## Abstract

Hyperglycemia-induced protein glycation and formation of advanced glycation end-products (AGEs) plays an important role in the pathogenesis of diabetic complications and pathological biomineralization. Receptors for AGEs (RAGEs) mediate the generation of reactive oxygen species (ROS) via activation of NADPH-oxidase. It is conceivable that binding of glycated proteins with biomineral particles composed mainly of calcium carbonate and/or phosphate enhances their neutrophil-activating capacity and hence their proinflammatory properties. Our research managed to confirm this hypothesis. Human serum albumin (HSA) was glycated with methylglyoxal (MG), and HSA-MG was adsorbed onto mineral microparticles composed of calcium carbonate nanocrystals (vaterite polymorph, CC) or hydroxyapatite nanowires (CP). As scopoletin fluorescence has shown, H_2_O_2_ generation by neutrophils stimulated with HSA-MG was inhibited with diphenyleneiodonium chloride, wortmannin, genistein and EDTA, indicating a key role for NADPH-oxidase, protein tyrosine kinase, phosphatidylinositol 3-kinase and divalent ions (presumably Ca^2+^) in HSA-MG-induced neutrophil respiratory burst. Superoxide anion generation assessed by lucigenin-enhanced chemiluminescence (Luc-CL) was significantly enhanced by free HSA-MG and by both CC-HSA-MG and CP-HSA-MG microparticles. Comparing the concentrations of CC-bound and free HSA-MG, one could see that adsorption enhanced the neutrophil-activating capacity of HSA-MG.

## 1. Introduction

Oxidative stress plays a major role in the development of diabetes complications, both microvascular and cardiovascular. Receptors for advanced glycation end-products (RAGEs) mediate the generation of reactive oxygen species (ROS) by various cells: endothelial cells of large and small vessels, monocytes/macrophages, and granulocytes [1]. RAGEs are pattern-recognizing receptors (PRR) participating in the innate immune response. RAGE signaling leads to the activation of NF-κB and NADPH-oxidase in phagocytes. In a diabetic setting, the high levels of AGEs caused by hyperglycemia lead to the hyperactivation of neutrophils and might contribute significantly to chronic inflammation in diabetes patients. Inflammation in turn triggers microcalcification [2,3]; thus, the AGE/RAGE cascade mediates vascular calcification oxidative stress mechanisms [2,4,5]. Endogenous mineral particles can activate macrophages [6,7] and neutrophils [8,9], inducing ROS and cytokine production. Biomineralization is characterized by the inclusion of proteins and other biomolecules into the calcifications [10], modulating their neutrophil-activating capacity [11]. Adsorption of the components of biological fluids should be also considered as an important factor in neutrophil–particle interaction. There are no data on the inclusion of glycated albumin into mineral particles and further neutrophil activation at present. It is conceivable that the binding of glycated proteins with biomineral particles composed mainly of calcium carbonate and/or phosphate enhances their neutrophil-activating capacity and hence their proinflammatory properties. Our research managed to confirm this hypothesis.

For this purpose, we adsorbed methylglyoxal-glycated human serum albumin (HSA-MG) onto high-porous vaterite 3 µm microspheres (CC) chosen as model microparticles and onto nanowire-composed hydroxyapatite microparticles (CP). Modification of albumin with MG is a complex reaction with multiple products, so first we analyzed its chemical properties and confirmed its activating effect towards neutrophils; then, adsorption of normal and modified albumin onto vaterite microparticles was assessed; and finally, the activation of neutrophils with vaterite microparticles coated with normal or MG-modified albumin was studied.

## 2. Results

### 2.1. Effects of Methylglyoxal-Induced Glycation of HSA

Methylglyoxal, formed by enzymatic and nonenzymatic routes from glycolytic intermediates and from the autoxidation of sugars, is known as a potent glycating agent, the production of which leads to the formation of AGEs. Methylglyoxal-, glyoxal- and glucose-modified AGEs are characterized by increased fluorescence and optical absorbance [12]. In order to confirm modification of HSA by MG in our experiments, we compared a number of HSA and HSA-MG parameters (Table 1).

As shown in Table 1, MG-induced HSA modification significantly increased protein fluorescence excited at 325 nm, with emission at 430 nm, which was consistent with the data of other researchers [12,13], and also absorbance at 320 nm [12,14]. According to mass spectrometry data, the modification also increased the *M*_W_ of the protein (Figure 1a,b). ζ-potential moved to negative values (Table 1), which was confirmed by the electrophoresis data (Figure 1c). The latter also revealed an increase in the *M*_W_ of HSA-MG compared to HSA: the *M*_W_ of a significant proportion of HSA-MG molecules was greater than those of HSA and BSA (Figure 1c). In addition, a fraction of protein with *M*_W_ 120 kDa was detected, which could have resulted from HSA-MG dimerization.

MG-induced modification did not influence Lowry assay results, which allowed us to use this method in further analysis.

### 2.2. Parameters of Protein Interaction with Vaterite Microparticles

The principal method of HSA/HSA-MG inclusion used in our study was adsorption onto vaterite microparticles in aqueous solutions [15] (CC-HSA and CC-HSA-MG), while in a separate experiment, hybrid protein–mineral particles were also prepared by coprecipitation (CC(HSA) and CC(HSA-MG)) [16]. Maximal adsorption of HSA-MG (7.4 ± 1.1 mg/g CC) was less than that of HSA (13.6 ± 2.5 mg/g CC). ζ-potential of CC, CC-HSA and CC-HSA-MG was 0 ± 0.5, −19.3 ± 0.7 and −33.9 ± 1.7 mV, respectively.

Protein inclusion by coprecipitation was higher than adsorption and gave 45 mg/g CC for CC(HSA) and 86 mg/g CC for CC(HSA); however, ζ-potential values corresponded to those for CC-HSA and CC-HSA-MG.

### 2.3. Neutrophil Activation by Soluble HSA-MG

The typical feature of AGE-induced neutrophil response is an increase in superoxide generation as a result of NADPH-oxidase activation [17]. The fluorometric assay based on decrease in scopoletin fluorescence as a result of H_2_O_2_-induced oxidation in the presence of HRP [18] is considered a reliable method for assessing NADPH-oxidase activity [19]. Scopoletin oxidation was minimal in the suspensions of neutrophils treated with native HSA (Figure 2a), indicating that HSA does not interfere with basal H_2_O_2_ production. In the presence of HSA-MG, scopoletin fluorescence steadily decreased (Figure 2a) as a result of increased H_2_O_2_ production by neutrophils (Figure 2b).

We used a fluorometric assay with scopoletin to identify signaling mechanisms participating in neutrophil NADPH-oxidase activation by HSA-MG (Figure 2c). EDTA was added to the probes as a chelating agent, and its inhibiting effect indicates the important role of cations, presumably calcium ions, in neutrophil activation to HSA-MG, which is consistent with the findings of [20] that AGE albumin evokes a transient increase in neutrophil calcium.

Diphenyleneiodonium chloride (DPI) is a well-known NADPH-oxidase inhibitor [19], so DPI-induced inhibition of scopoletin oxidation confirms a key role of NADPH-oxidase in neutrophil response to HSA-MG. Wortmannin is an inhibitor of neutrophil respiratory burst activated via G-coupled receptors or protein tyrosine–kinase coupled receptors. Both pathways depend on phosphatidylinositol 3-kinase (PIK3) activity, inhibited by wortmannin. Genistein, an inhibitor of protein tyrosine kinases, also blocked HSA-MG-induced scopoletin oxidation.

HSA-MG-induced increase in superoxide production by neutrophils was assayed also by Luc-CL. We detected an increase in Luc-CL of neutrophils stimulated with HSA-MG but not with HSA (Figure 3).

The effects of HSA-MG were dose-dependent and gave higher Luc-CL values than HSA in the tested range of concentrations (Figure 4).

As one can see, the data for the Luc-CL measurement were consistent with the results of the fluorometric assay, indicating NADPH-dependent stimulation of superoxide production by neutrophils induced by HSA.

We also examined HSA-MG-stimulated reaction of blood cells without separation. Luc-CL of diluted blood was not sensitive to HSA-MG, and addition of the modified protein up to 2 mg/mL had no effect on spontaneous CL levels.

It is known that erythrocytes significantly affect neutrophil response via Siglecs [21] and thus could prevent HSA-MG-induced effects on neutrophil NADPH-oxidase.

### 2.4. Neutrophil Activation by Protein–Vaterite Microparticles

As was shown in our experiments with unbound HSA-MG, Luc-CL is a sensitive and adequate method for assessment of neutrophil NADPH-oxidase activation.

Vaterite microparticles are biocompatible and nontoxic and did not significantly activate neutrophil Luc-CL, as with CC-HSA microparticles, unlike CC-HSA-MG (Figure 5a). The effect depended on the quantity of adsorbed HSA-MG (Figure 5b) and on particle concentration (Figure 5c). The ratio of particles to neutrophils ranged from 2:1 to 20:1, which corresponded to 0.1–1.0 mg/mL of the particles, while neutrophil concentration was 0.5 × 10^6^ cells/mL.

In spite of the greater protein inclusion in microparticles fabricated by coprecipitation, their neutrophil-stimulating activity was close to that of vaterite with adsorbed proteins and higher than that of co-precipitation with HSA microparticles (Figure 5d).

In experiments with neutrophils from four healthy volunteers, we also compared the stimulation of neutrophils with untreated microspheres (CC), microspheres opsonized with human autologous serum (CC-ops) and microspheres coated with HSA (CC-HSA) vaterite and found that there was no significant difference between these samples: the Luc-CL amplitudes were 7.1 ± 1.7 V (CC), 9.1 ± 1.2 V (CC-ops), 8.0 ± 1.7 V (CC-HSA), respectively.

It should be noted that to reach the Luc-CL that exceeded spontaneous values by more than two times, the HSA-MG concentration in the solution was more than 0.1 mg/mL (Figure 4), while for vaterite-bound HSA-MG it was 1 µg/mL (Figure 5b).

The microparticle-stimulated Luc-CL response of neutrophils dropped in the presence of 120 Un/mL SOD, as shown in Figure 6. SOD was added into the probes after the maximal CL value was reached and the effect was calculated as CL_2_/CL_1_ × 100%.

In the tested range of HSA-MG concentration of 2–6 µg/mL as adsorbed protein, SOD decreased CL by 36 ± 3% up to 74%. This means that Luc-CL registered mainly superoxide radical generation.

Hydroxyapatite is another important biomineral found in various calcifications, along with calcium carbonate. We used nanostructured microparticles composed of nanowires (CP, Figure 7b) to see if they activate neutrophils after HSA-MG adsorption. Although the CP surface properties need to be studied further, 12 mg/g HSA-MG significantly enhanced neutrophil Luc-CL response (Figure 8).

This result demonstrates that the precipitation of phosphate ions onto calcium carbonate particles which takes place in biological fluids [22,23] would not abolish eventual HSA-MG inclusion and neutrophil activation.

## 3. Discussion

Mineral–organic particles containing calcium phosphate and proteins, such as albumin, fetuin-A and apoliprotein-A1, were detected in calcified arteries [24]. Mineral deposition in vivo occurs due to mineral–organic nanoparticles containing blood proteins readily binding to various organic molecules in body fluids [25].

Neutrophils, bone marrow-derived innate immune cells, are among the first inflammatory cells in the host response to infection and other danger signals. Phagocytosis of cholesterol, bilirubin, calcium hydroxyapatite and calcium carbonate crystals by human neutrophils is accompanied by ROS production, indicating the proinflammatory role of these interactions [6,8]. Calcium phosphate-based microparticles fabricated by co-precipitation with BSA or fetuin-A elicited intracellular ROS production, which was abolished by NADPH-oxidase inhibitors, and the formation of neutrophil extracellular traps (NETs). Moreover, activation of neutrophils stimulated also macrophages proinflammatory reaction [11,26].

Similarly, activation of macrophages with calcium phosphate crystals measuring less than 1 µm resulted in the release of TNF-α, IL-1β and IL-8 [26], as well as CaCO_3_-based particles (needle-shaped aragonite and phosphate-coated aragonite measuring 15–20 µm), which induced THP1 macrophage release of TNF-α and IL-8 [7].

The neutrophil-activating role of the proteins included in mineral–organic particles is still unclear. The cellular effects of calcium phosphate crystals are well known to be modulated by proteins adsorbed at the crystal surface. Thus, adsorption of apolipoprotein B on crystals abrogated their inflammatory potential, whereas adsorption of immunoglobulin G (IgG) increased their inflammatory activity [27,28].

One could hypothesize that the adsorbed or co-precipitated glycated proteins bound with the particles might be ligands for neutrophil RAGEs and thus might activate cellular NADPH-oxidase and ROS generation; our purpose was to study effects of absorbed HSA-MG on neutrophils. We also compared the effects of vaterite adsorbed and coprecipitated with HSA and HSA-MG.

To prepare AGEs, we modified HSA with methylglyoxal within 7 days, and the resulting products had typical AGE fluorescences and optical absorbances [12].

Mass spectrometry showed a maximal increase in HSA-MG by ~7900 Da, which was almost twice the value of 4500 Da reported by Mera [29]. According to DIGE and MALDI, HSA-MG was a mixture of modified HSA molecules ranging in *M*_W_ from 66 kDa to 76 kDa, and dimers were detected by DIGE with *M*_W_ ~120 kDa. Methylglyoxal glycation also increased the net negative charge of the proteins, as was shown by ζ-potential assay and by DIGE, which is consistent with other researchers’ data [29].

Our study included experiments with high-porous vaterite microspheres (CC) measuring ~3 µm in diameter as model microparticles, and in some experiments nanowire-composed hydroxyapatite microparticles (CP) of ~40 µm in diameter were also tested. CC microspheres are well known for their capacity to bind various proteins [23,30,31,32].

Adsorption of HSA and HSA-MG onto vaterite microparticles (CC) differed significantly, with maximal values for HSA which were 2-fold less than those for HSA-MG. This could be attributed to the difference in *M*_W_ and hence molecular size between the intact and modified proteins. Unlike adsorption, co-precipitation resulted in greater HSA-MG inclusion compared to HSA. To assess neutrophil activation by CC-HSA and CC-HSA-MG microparticles, we used Luc-CL, which is a sensitive method for detection of extracellular superoxide anion generation [33]. Indeed, in our experiments SOD decreased Luc-CL stimulated with CC-HSA-MG by 70%. Moreover, H_2_O_2_ production by neutrophils activated with HSA-MG was reduced by the NADPH-oxidase inhibitor DPI, according to scopoletin fluorescence assay. RAGEs are known to activate NF-κB via the MAPK–Erk1/Erk2-dependent pathway [34]. The inhibitory effects of wortmannin and genistein indicated that the HSA-MG-stimulated ROS production by neutrophils depended on PI3K and protein tyrosine kinase. Previously, it was shown that PI3K activation in neutrophils mediated neutrophil adhesion and migration in AGE-collagen [35].

When blood was incubated with HSA-MG, no increase in Luc-CL was detected, presumably because of erythrocyte-dependent modulating effects [21]. This result might indicate that systemic HSA-MG-induced activation of neutrophils in blood is unlikely.

One of the most interesting findings in our study was the difference in neutrophil-activating capacity between free and coated microparticles of HSA-MG. Indeed, a total 1–5 µg/mL CC-bound HSA-MG was as effective as 100–500 µg/mL free protein, as assayed by Luc-CL. Large, positively charged patches on RAGE V and C1 domains are traps for negatively charged ligands [36], and receptor oligomerization on plasma membranes was registered [37]. Initiation of signal cascades by ligand-induced oligomerization is one of the possible mechanisms of RAGE activation [37], and immobilization of glycated protein on the particles could favor this activation process. Moreover, the co-stimulation of neutrophils with AGEs and mineral components is not implausible: crystals of triclinic calcium pyrophosphate dihydrate activated MAP kinase in neutrophils [38].

In this study, we did not focus on effects of protein oligomerization, even though some HSA-MG dimers were detected by electrophoresis. Oligomerization can be accompanied by significant conformational changes which are significant for RAGE-dependent neutrophil responses. Thus, lately it was shown that modification of BSA influences the stiffness and formation of β-rich fibrils [39], which are well-known RAGE agonists [1].

Our results suggest that HSA-MG–RAGE interaction could induce NADPH-oxidase activation in neutrophils, and we suppose that the same mechanism is operative in the reaction of neutrophils to CC-HSA-MG. The method of HSA-MG inclusion—adsorption or co-precipitation—did not significantly influence the resultant superoxide production. Moreover, hydroxyapatite microparticles composed of nanowires (CP) coated with HSA-MG also activated neutrophil Luc-CL in spite of differing significantly from CC in terms of their size and structure. Thus, the interaction of biominerals varying in nature and size with HSA-MG (and possibly with other glycated proteins) can be considered a potent proinflammatory factor.

## 4. Materials and Methods

### 4.1. Reagents

CaCl_2_, ≥93.0%; Na_2_CO_3_, ≥99.0%; Histopaque 1.077, 1.119; Krebs–Ringer solution; Folin–Ciocalteu reagent; human serum albumin fraction V (HSA); methylglyoxal; lucigenin; superoxide dismutase (SOD); horseradish peroxidase (HRP); sodium azid (NaN_3_); scopoletin; trisodium citrate; genistein; wortmannin; diphenyleneiodonium chloride (DPI); ethylenedinitrilotetraacetatic acid (EDTA); Tris; 3-((3-cholamidopropyl) dimethylammonio)-1-propanesufonate (CHAPS); nonyl phenoxypolyethoxylethanol (NP40); *N*,*N*,*N*′,*N*′-tetramethyl ethylenediamine (TEMED); ammonium persulfate; glycerol; sodium dodecyl sulfate (SDS); thiourea; acrylamide; and bisacrilamide were all purchased from Sigma-Aldrich (St. Louis, MO, USA); glycine was purchased from ApliChem GmbH (Darmstadt, Germany); gelatin from Fluka (Seelze, Germany) (; dextran T70 from Roth, Germany; USP-grade urea from Amresco (Solon, OH, USA); 2,5-dihydroxybenzoic acid from Bruker Daltonics (Billerica, MA, USA); cyanines from Lumiprob (Hunt Valley, ML, USA); ampholines, DIGE standards, dithiothreitol (DTT) from BioRad (Hercules, CA, USA); and NaH_2_PO_4_∙2H_2_O, propanol-2, CuSO_4_·5H_2_O, NaCl, Eur Ph grade CaCl_2_·2H_2_O and sodium citrate were purchased from “Chimmed”(Moscow, Russia).

### 4.2. Preparation of Methylglyoxal-Modified Albumin (HSA-MG)

To prepare HSA-MG, 10 mg/mL HSA was incubated with 100 mM MG at 37 °C for up to 7 days in 50 mM borate buffer solution (pH8.6). The reaction mixture was then diluted with a 15-fold excess of Krebs–Ringer solution and concentrated using an Amicon *ultra* centrifugal filter with a membrane NMWL of 3 kDa to remove MG; the washing was repeated thrice. The resultant HSA solution was stored in aliquots at −70 °C. The final protein concentration was assayed by Lowry’s method.

### 4.3. The Molecular Mass of HSA and HSA-MG

Prior to analysis on a mass spectrometer, the reaction mixture was desalted using a Millipore ZipTip C-18 according to a protocol recommended by the manufacturer. The sample was eluted from the tip with 2 μL of 30% acetonitrile–DI water (*v*/*v*) and then mixed with 2 μL of MALDI matrix solution (2,5 Dihydroxybenzoic acid). The 1 μL aliquot was than spotted on the MALDI sample plate and air-dried at ambient temperature.

Mass spectroscopic analysis was carried out with a MALDI–TOF (Matrix-Assisted Laser Desorption Ionization–Time-of-Flight Mass Spectrometry) device—the Bruker Ultraflex II (Bruker, Bremen, Germany). Spectra were recorded in the linear mode for positive ions at 25 kV accelerating voltage. For each spectrum, data from 200–400 laser shots were accumulated.

### 4.4. Advanced Glycation Product (AGE) Detection

To detect the methylglyoxal-induced formation of AGEs, fluorescence of HSA-MG and HSA was monitored by exciting the samples at 325 nm with emissions at 350–500 nm using a computerized spectrofluorometer—the SOLAR SM 2230 (SOLAR, Minsk, Belarus)—and a 10 mm path length quartz cell. UV–visible spectra (in the range of 230–500 nm) of the protein solutions were registered using a UV spectrophotometer (Varian Cary 50 Bio, Varian Australia Pty. Ltd., Melbourne, Victoria, Australia).

### 4.5. Two-Dimensional Difference Gel Electrophoresis (DIGE)

Before DIGE electrophoresis [40], the HSA and HSA-MG solution (10 mg/mL) was diluted in 40 mM Tris-HCl (pH 9.5) buffer containing 8 M urea, 2 M thiourea, 4% CHAPS+NP40. The samples were centrifuged at 14,000× *g* for 15 min. Protein concentration in the samples was measured by the Bradford method using Quick Start Bradford dye (BioRad). The sample proteins were labeled with Cy3 (green) or Cy2 (blue) CyDye DIGE dyes (Lumibrobe (Moscow, Russia)) according to the manufacturer’s instruction (400 pmol for 50 μg protein), and DIGE standards (BioRad) were labeled with Cy5 (red). After stopping the binding reaction of cyanines with protein by 10 mM lysine solution, DTT was added to a final concentration of 100 mM and Ampholine 3,5-10 (Bio-Rad) to 1%. Before mixing the two compared samples, we performed electrophoretic separation of each of them by electrophoresis on 12% PAAG under denaturing conditions. After electrophoresis, the gel was scanned on a TyphoonTrio scanner, Amersham (Marlborough, MA, USA) at a laser wavelength of 532 nm (green fluorescence), 488 nm (blue fluorescence), and the value of the total fluorescence intensity for each sample was defined. At these values, two samples (HSA and HSA-MG) labeled with different cyanines were mixed in a certain ratio, based on the general equalization of the fluorescence intensity values for each of them. Isoelectrofocusing was performed in 18 cm glass tubes in 4% polyacrylamide gel (8 M urea, 2% ampholines (pH 3.5–10) and 4% ampholines (pH 5–7), 6% solution containing 30% CHAPS and 10% NP 40, 0.1% TEMED, 0.02% ammonium persulfate). The total protein content was 200–250 μg in the tubes. Isoelectrofocusing was performed in the following mode: 100, 200, 300, 400, 500 and 600 V, for 45 min; 700 V for 10 h; 900 V for 1 h. On completion of isoelectrofocusing, the tubes were equilibrated in 62.5 mM Tris-HCl buffer (pH 6.8) containing 6 M urea, 30% glycerol, 2% SDS, 20 mM DTT and bromophenol blue, for 30 min. Then, the tubes were transferred to the surface of gradient polyacrylamide gel (9–16%) and fixed with 0.9% agarose with bromophenol blue. Electrophoresis was performed in Tris–glycine buffer under cooling in the following mode: 20 mA on glass, for 20 min; 40 mA on glass, for 2 h; 35 mA on glass, for 2.5 h, under chamber cooling to 10 °C. The gels were scanned on a Typhoon Trio scanner (Amersham) at 532 nm (Cy3), 488 nm (Cy2) and 633 nm (Cy5), at a laser intensity of 500 pmt.

### 4.6. Scanning Electron Microscopy (SEM)

Immediately before sample deposition, silicon wafers were treated in plasma cleaner Electronic Diener (Plasma Surface Technology, Ebhausen, Germany). The CC or CPNW particles were then deposited onto them, covered with a 10 nm layer of Au–Pd using the Sputter Coater Q150T (Quorum Technologies, Lewes, UK) and characterized using a Zeiss Merlin microscope equipped with GEMINI II Electron Optics (Zeiss, Oberkochen, Germany). The SEM parameters were: accelerating voltage: 1–3 kV; and probe current: 80–300 pA.

### 4.7. Fabrication of Vaterite Microparticles and Vaterite Microparticles with Co-Precipitated Protein

Vaterite microparticles (CC) were synthesized as described previously [16]. The mixture of 9 mL of 0.05 M Tris buffer (pH 7.0) with 0.3 mL 1 M CaCl_2_ (pH 7.0) and 3 mL 0.1 M Na_2_CO_3_ was stirred at RT, and the formed crystals were washed twice with pure water and lyophilized. For the fabrication of hybrid microparticles with mucin, 3 mL of 1 M CaCl_2_ containing 8.3 mg mL^−1^ of mucin in 0.05 M Tris buffer (pH 7.0) was stirred for 10 min followed by the addition of 3 mL of 1 M Na_2_CO_3_ water solution. The precipitate was separated by centrifugation for 2 min at 1000× *g*, washed twice with double-distilled water and lyophilized (Figure 7a).

Vaterite–protein microparticles with co-precipitated protein were prepared by the method of coprecipitation using 4 mg/mL HSA or HSA-MG in 0.05 M Tris buffer (pH 7.0), as described above. To assess protein inclusion, optical absorbance at 280 nm was controlled in supernatants and washing solutions at all stages.

The characteristics of CC were consistent with our previous data: 3.3 ± 0.8 µm in diameter, with nanocrystals of 109 nm. As shown by the BET method, the surface area of the CC particles was 4.3 m^2^/g [16].

### 4.8. Light Microscopy of Vaterite Microparticles (CC)

To calculate vaterite particles in suspension, a light Motic BA223 microscope (Motic, Hong Kong) equipped with a 3CCD KYF32 digital camera was used. Image processing was performed with a MECOS-C image analysis system (MECOS, Moscow, Russia) in semi-automatic mode (400×). Cell concentration was assayed by direct counting using a Goryaev chamber.

### 4.9. Calcium Phosphate Nanowire-Composed Microparticles (CP) Preparation

CP particles were synthesized according to modified procedure of Zhan et al. [41]. Briefly, 0.25 g of gelatin, 0.735 g of CaCl_2_∙2H_2_O and 0.6 g of urea were dissolved in 0.2 L of warm double-distilled water (ddH_2_O) and the temperature was raised to 100 °C. Then, 0.78 g of NaH_2_PO_4_∙2H_2_O, dissolved in 50 mL of ddH_2_O, was added dropwise, and the reaction mixture was stirred at 85 °C for 3 days. The collected precipitate was then centrifuged, washed three times with IPA and dried in vacuum at 60 °C to constant weight. According to SEM, most of the nanowires were 2–20 µm long and 30–200 nm thick, although bundles of up to 500 nm were also present (Figure 7b).

### 4.10. Protein Adsorption onto CC and CP Particles

Suspensions of 10 mg mL^−1^ CC and CP particles in 0.15 M NaCl were separately mixed with an equal volume of 10 mg mL^−1^ protein solution (unless otherwise indicated) and incubated for 30 min at 37 °C under periodic shaking. The precipitates were separated by centrifugation for 10 min at 1000 g, washed twice with 0.15 M NaCl and resuspended to 10 mg mL^−1^, giving CC-HSA, CC-HSA-MG, CP-has and CP-HSA suspensions. The supernatants and washing solutions were collected for further protein concentration assays according to Lowry’s method.

### 4.11. ζ-Potential Measurement

The ζ-potential of microparticles and proteins was measured using a Zetasizer (Nano ZS, Malvern, Oxford, UK) and estimated using Smoluchowski eq.

### 4.12. Blood Collection and Isolation of Neutrophils

Normal blood of 5 healthy volunteers was collected with their informed consent and agreement and stored in EDTA vacutainers. Aliquots of whole blood were used immediately in a chemiluminescence assay. Another blood volume was layered over the double gradient of Histopaque 1.077/1.119 g L^−1^ and, after centrifugation for 45 min, neutrophils were collected and washed with Krebs–Ringer solution. Cell concentration was assayed by direct counting using a Goryaev chamber.

For fluorometric assays, to avoid the influence of EDTA, the blood was collected into tubes containing 109 mM trisodium citrate as an anticoagulant at a ratio of 9:1. Neutrophils were isolated by centrifugation in the histopaque-1077 density gradient, as previously described [42,43].

### 4.13. Measurement of H_2_O_2_ Production by Human Neutrophils Based on Scopoletin Oxidation

H_2_O_2_ production by neutrophils was measured using the scopoletin–horseradish peroxidase (HRP) fluorescence technique [44]. Briefly, a suspension of neutrophils (2 × 10^6^ cells/mL in PBS supplemented with 1 mM CaCl_2_ and 0.5 mM MgCl_2_) was mixed with 1 μM scopoletin (a fluorescent substrate of HRP), 20 μg/mL HRP and 1 mM NaN_3_ (catalase and myeloperoxidase inhibitor). The cell suspension was incubated for 5 min at 37 °C in a cuvette of a spectrofluorometer SM 2203 (SOLAR, Minsk, Belarus) and then test reagents were added as required. A decrease in the fluorescence of scopoletin was monitored at 460 nm (excitation at 350 nm) and the maximal slope of the recorded traces was calculated and used to quantify the rate of H_2_O_2_ generation by cells.

### 4.14. Chemiluminescence Assay (CL)

Lucigenin-enhanced CL (Luc-CL) of isolated neutrophils was measured with the Lum1200 luminometer (DiSoft, Moscow) in 0.5 mL of Krebs–Ringer solution (pH 7.4) with 0.1 mM lucigenin (Luc-CL), 2% autologous blood serum, 0.5–0.7 × 10^6^ mL^−1^ neutrophil cells and 1 mg mL^−^^1^ particles. Spontaneous CL was measured before the addition of particles, then the sample was added and CL was registered until maximum values were reached; the CL amplitude (V) was calculated as the difference between maximum and spontaneous values. If necessary, superoxide dismutase (SOD) was added just when the maximum was achieved.

CL of the whole blood CL was assayed by the same technique, with 25 µL of blood finally diluted at a ratio of 1:25.

## Figures and Tables

**Figure 1 ijms-23-07840-f001:**
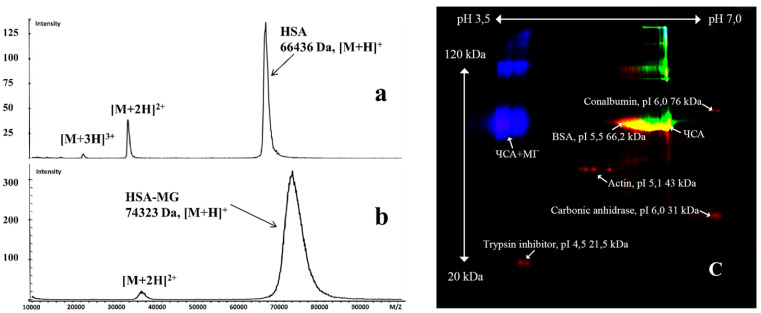
Molecular mass of HSA (**a**) and HSA-MG (**b**), analyzed by a MALDI–TOF mass spectrometer, as described in the Materials and Methods section. (**c**) DIGE analysis of HSA (green) and HSA-MG (blue) (pH range: 3.5–7).

**Figure 2 ijms-23-07840-f002:**
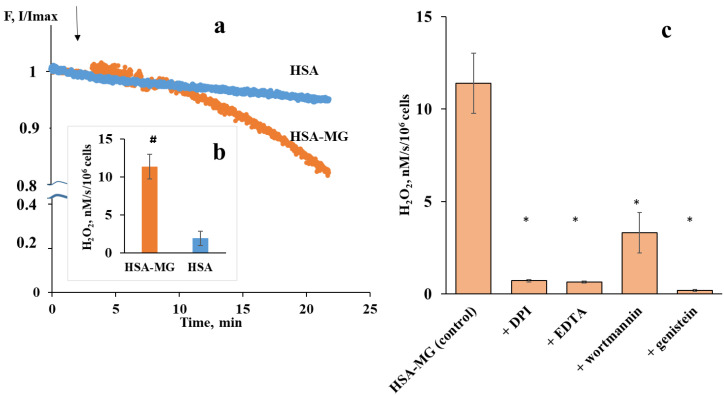
Typical kinetic curves of scopoletin oxidation by human neutrophils as a measure of H_2_O_2_ generation in response to native HSA or HSA-MG (**a**); H_2_O_2_ production by neutrophils as the rate of scopoletin oxidation (**b**); and the effects of DPI, EDTA, genistein and wortmannin on the rate of scopoletin oxidation by neutrophils in response to HSA-MG (**c**). DPI was used at a concentration of 20 μM; EDTA: 1 mM (experiments with EDTA were performed in PBS free of CaCl_2_ and MgCl_2_); genistein: 50 μM; and wortmannin: 100 nM. Cells were incubated with inhibitors or EDTA for 5 min at 37 °C, then HSA-MG was added. The concentration of HSA and HSA-MG was 25 µg/mL. Data are presented as the means ± SEMs, *n* = 5–6. ^#^ *p* < 0.05 vs. HSA; * *p* < 0.05 vs. the effect of HSA-MG.

**Figure 3 ijms-23-07840-f003:**
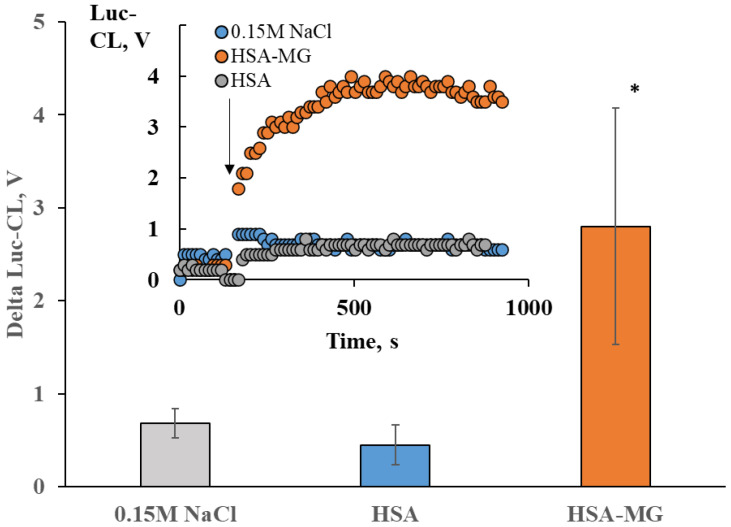
Luc-CL neutrophil response to 1 mg/mL HSA or HSA-MG. The inset shows typical kinetic curves for Luc-CL (V). The bars show the amplitude of the Luc-CL assay performed in triplicate. * *p* < 0.05 vs. HSA (according to Student’s *t*-test).

**Figure 4 ijms-23-07840-f004:**
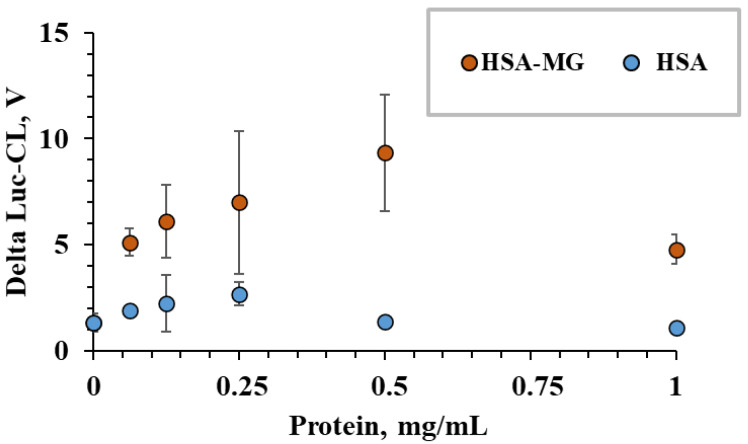
Dose-dependent stimulation of neutrophil Luc-CL by HSA and HSA-MG. Each point represents the amplitude of neutrophil CL response measured in triplicate as mean value and standard deviation.

**Figure 5 ijms-23-07840-f005:**
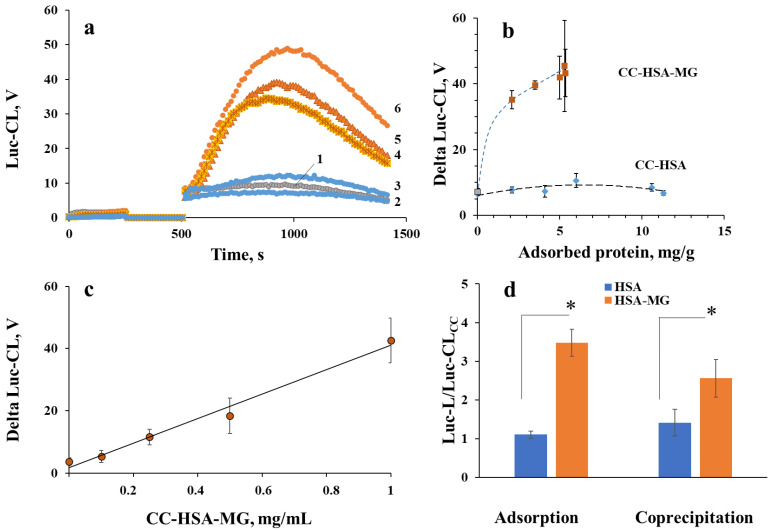
Luc-CL of neutrophils, stimulated by CC with adsorbed proteins: kinetics (**a**) for CC without proteins (curve 1), with 11 mg/g and 6 mg/g HSA (curves 2 and 3), and with 2.1 mg/g, 3.5 mg/g, and 5 mg/g HSA-MG (curves 4, 5 and 6). Amplitude (**b**–**d**) is presented as a function of the quantity of adsorbed protein (**b**), of the concentration of CC-HSA-MG (with protein inclusion of 6 mg/g) (**c**) and, in comparison, with CC, coprecipitated with HSA and HSA-MG (**d**). * *p* < 0.05 vs. CC-HSA (according to Student’s *t*-test).

**Figure 6 ijms-23-07840-f006:**
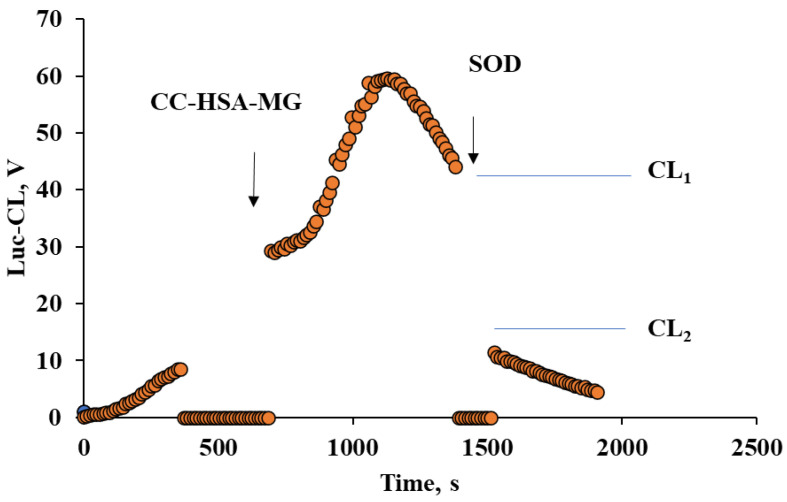
Kinetics of Luc-CL of neutrophils stimulated by vaterite with adsorbed HSA-MG (5 mg/g CC) before and after 120 Un/mL SOD addition. Microparticle concentration: 1 mg/mL; neutrophil concentration: 0.5 × 10^6^ cells/mL.

**Figure 7 ijms-23-07840-f007:**
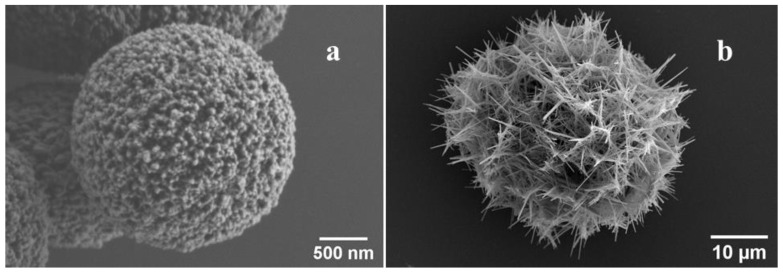
Scanning electron image of typical vaterite microparticle (**a**) and a microparticle composed of calcium phosphate nanowires (**b**).

**Figure 8 ijms-23-07840-f008:**
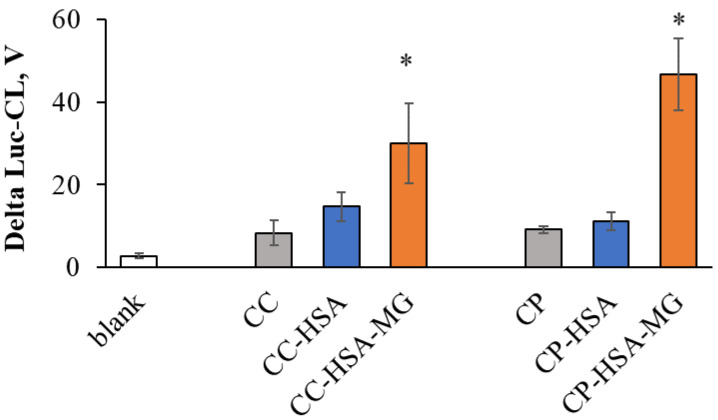
Amplitude of neutrophil Luc-CL, stimulated by vaterite (CC) or hydroxyapatite (CP) microparticles before and after coating with HSA or HSA-MG. Microparticle concentration: 1 mg/mL, neutrophil concentration: 0.5 × 10^6^ cells/mL. As a blank control, 0.15 M NaCl solution was added to neutrophils. Each bar represents the mean value of three independent experiments. * *p* < 0.05 vs. the same untreated microparticles or microparticles coated with HSA, according to Student’s *t*-test.

**Table 1 ijms-23-07840-t001:** Characteristics of HSA and HSA-MG.

Parameter	HSA	HSA-MG	*p*
Fluorescence, λ_ex_ = 325 nm/λ_em_ = 430 nm	0.31 ± 0.06 *	0.41 ± 0.02 *	˂0.001
Absorbance, λ = 320 nm	0.016 ± 0.001 *	1.376 ± 0.002 *	˂0.001
ζ-potential, mV	−13.8 ± 2.2	−15.9 ± 2.0	
*M*_W_, Da	67,626	75,461	

* Protein concentration: 1 mg/mL.

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
