# Peer review of "Neutrophil Activation by Mineral Microparticles Coated with Methylglyoxal-Glycated Albumin"

_ijms, 2022, doi:10.3390/ijms23147840_

Round 1

Reviewer 1 Report

The manuscript by Mikhalchik et al. investigated the effects of mineral microparticles coated with methylglyoxal-glycated albumin on neutrophil activation. The authors found that compared to unmodified albumin, methylglyoxal-glycated albumin increased superoxide generation by neutrophils. Moreover, the authors also showed that coating two different types of microparticles with the modified protein further enhanced the activation of neutrophils compared to the free forms of the modified protein. The manuscript is concise and clear, and the authors should be commended for their study. However, there are a few issues that the authors need to address:

Major concerns:

  1. In the abstract, the authors mentioned that their study tested the hypothesis that “binding of glycated proteins with biomineral particles composed mainly of calcium carbonate and/or phosphate enhances their neutrophil-activating capacity.” There are two components here:

i.                    The role of biomineral particles and

ii.                  The role of glycation as a protein modification.

To test the specificity of the biomineral microparticles, the authors need to use an inert particle (such as latex beads) as their negative/ specificity control. The authors should coat the latex beads with the same modified protein (HSA-MG) and compare their effects on neutrophil activation. Instead of comparing CC-HAS vs. CC-HAS-MG, the authors should compare CC-HAS-MG vs. Latex-HAS-MG. Without this control, the conclusions about the roles of biomineral microparticles cannot be validated. This is especially important as the authors concluded that “One of the most interesting findings in our study is the difference in neutrophil-activating capacity between free and coating microparticles HSA-MG.” Without this control, it is impossible to discern if this results from the coating on biomineral particles or if any particles with coated modified proteins would produce the same effects.

To test the role of glycation, the authors compared modified protein with unmodified protein. How about a different modification? Can the same results be achieved when a random protein modification, such as nitrosylated or even ubiquitinated protein, is used?

2. What is the purpose of whole blood stimulation? On the one hand, the authors found that adding a large amount of modified protein had no effects on superoxide generation (Fig 9). On the other hand, the authors showed an increase in plasma elastase concentration. The authors need to explain this discrepancy. The conclusion that “The effect of HSA-MG on neutrophil degranulation could be mediated by other blood cells and/or proteins and needs further investigation” doesn’t clarify anything; rather, it brings in more questions. If the erythrocytes affected neutrophil response (as the authors correctly pointed out), then why is there an increase in elastase secretion but not superoxide generation in the presence of erythrocytes? If further studies are required, why do the authors need to show something unrelated to their research question?

3. There is an apparent lack of focus: the title and abstract mention neutrophil activation by mineral microparticles coated with methylglyoxal-glycated albumin. However, the first time the authors showed the effect of coated microparticles was in figure 11. The first ten figures show the chemical aspects, the effects of modified vs. unmodified proteins, and the effects on whole blood stimulation. These could easily be combined into two or three figures: the chemical part can be put in a supplementary file, and the whole blood stimulation can even be omitted. It is hard to justify the title and abstract from the organization of the text.

 Minor concerns:

  1. The authors showed that compared to unmodified HSA, HSA-MG produced a statistically significant increase in superoxide generation by neutrophils. It will be interesting to know how this increase compares with that obtained by a classical RAGE-agonist, such as HMGB1.
  2. Why is Figure 13 suddenly showing 3D graphs with no error bars? The legend should mention the “n.”
  3. There are many typos or incomplete information. For example, Fig 7 legend says, “На вставке нет подписи на оси Y.” Author contribution shows “The following statements should be used “Conceptualization, X.X. and Y.Y.; methodology, X.X.; software, X.X.; validation, X.X., Y.Y. and Z.Z.; formal analysis, X.X.; investigation, X.X.; resources, X.X.; data curation, X.X.; writing—original draft preparation, X.X.; writing—review and editing, X.X.; visualization, X.X.; supervision, X.X.; project administration, X.X.; funding acquisition, Y.Y. All authors have read and agreed to the published version of the manuscript.”

Reviewer 2 Report

The authors have compared the effects of human serum albumin (HSA), methylglyoxal-glycated albumin (HSA-MG), calcium carbonate nanocrystals coated with HSA or HSA-MG, and hydroxyapatite nanowires with HSA or HSA-MG on the neutrophil oxidative burst (O2- and H2O2). They report on the activation of the oxidative burst by HSA-MG, possibly through AGE receptor-mediated signalling. Pharmacologic inhibition of tyrosine kinases, phosphatidylinositol 3-kinase or calcium chelation reduced HSA-MG-mediated activation of the respiratory burst in neutrophils. Furthermore, calcium carbonate nanocrystals and hydroxyapatite nanowires coated with HSA-MG were far more potent at activating the NADPH oxidase when compared to HSA-MG in solution.

General comment. This is a very descriptive study. The mechanistic approach is weak and would require additional experiments to assess why calcium carbonate nanocrystals or hydroxyapatite nanowires coated with HSA-MG strongly stimulate the oxidative burst compared to HSA-MG.

Major points:

1. There are a lot of unnecessary details. Information on the characterization of calcium carbonate and hydroxyapatite microcrystals can be provided as supplemental information or data.

2. The study rationale is that adsorption of the HSA-MG to various particles promotes neutrophil-dependent activation and inflammation. It is likely true, but there is insufficient focus on this novel and most interesting point. 

3. In whole blood, HSA-MG was unable to stimulate NADPH oxidase but enhanced the release of O2- when added to PMA-stimulated neutrophil. The data are not convincing since NaCl (the vehicle?), and HSA induce a small oxidative burst. I somewhat disagree with the author’s suggestion regarding the modulating effect of erythrocytes on the HSA-MG-mediated oxidative burst. Blood dilution (1:25) may reduce the number of neutrophils in the assay to a level that does not allow the detection of O2-.

4. Regarding the signalling mechanisms, the conclusions only apply to HSA-MG. The calcium carbonate nanocrystals or hydroxyapatite nanowires coated with HSA-MG likely stimulate additional signalling pathways. For example, activation of phosphatidylinositol 3-kinase is dependent on the structure and the microparticles.

Minor points:

Figure 6: The figure legend should include the concentration of HSA-MG.

Are HSA-MG monomers and HSA-MG dimers equally potent in stimulating the neutrophil oxidative burst?

Round 2

Reviewer 1 Report

The authors addressed most of the issues. However, just metioning the data is not sufficient. This is an important control, and need to be shown in the figure (or at least in a supplementary figure).

"In our preliminary experiments, we also compared stimulation of normal neutrophils with untreated and opsonized with human autologous serum vaterite microspheres, and no significant effect was detected compared with HSA-coated microparticles (data not shown)."
